# ISIDOG Recommendations Concerning COVID-19 and Pregnancy

**DOI:** 10.3390/diagnostics10040243

**Published:** 2020-04-22

**Authors:** Francesca Donders, Risa Lonnée-Hoffmann, Aristotelis Tsiakalos, Werner Mendling, José Martinez de Oliveira, Philippe Judlin, Fengxia Xue, Gilbert G. G. Donders

**Affiliations:** 1Femicare VZW Clinical Research for Women, 3300 Tienen, Belgium; francesca.donders@gmail.com; 2Department Gynecology, Hospital St Olav, 7010 Trondheim, Norway; Risa.Lonnee-Hoffmann@stolav.no; 3Department Ob/Gyn, LETO Obstetrician Gynecological & Surgical Center, 11525 Athens, Greece; atsiakalos@gmail.com; 4German Center for Infections in Gynecology and Obstetrics, 42283 Wuppertal, Germany; w.mendling@t-online.de; 5Department OB/Gyn, University Interior Beira, 6200 Covilha, Portugal; jmo@fcsaude.ui.pt; 6Department OB/Gyn, CHU De Nancy—Université de Lorraine, 54000 Nancy, France; pjudlin@gmail.com; 7Department OB/Gyn, Tianjin Medical University General Hospital, Tianjin 30000, China; dongmengting@gmail.com; 8Department Ob/Gyn, University Hospital Antwerp, 2650 Ekeren, Belgium

**Keywords:** coronavirus, COVID-19, maternal complications, pandemic, obstetric complications, pregnancy outcome, review

## Abstract

Providing guidelines to health care workers during a period of rapidly evolving viral pandemic infections is not an easy task, but it is extremely necessary in order to coordinate appropriate action so that all patients will get the best possible care given the circumstances they are in. With these International Society of Infectious Disease in Obstetrics and Gynecology (ISIDOG) guidelines we aim to provide detailed information on how to diagnose and manage pregnant women living in a pandemic of COVID-19. Pregnant women need to be considered as a high-risk population for COVID-19 infection, and if suspected or proven to be infected with the virus, they require special care in order to improve their survival rate and the well-being of their babies. Both protection of healthcare workers in such specific care situations and maximal protection of mother and child are envisioned.

## 1. Introduction

These recommendations are based on currently available published or in-press peer reviewed case studies of COVID-19 and pregnancy (English only) and on the following guidelines: Centers for Disease Control and Prevention (CDC) guidelines [1], Royal College of Obstetrics and Gynaecology (RCOG) guidelines [2] and Australian and New Zealand Intensive Care Society (ANZICS) guidelines [3].

Currently little is known about the exact management of pregnant women in COVID-19 endemic settings. Based on an extensive literature review, the International Society of Infectious Disease in Obstetrics and Gynecology (ISIDOG) provides herewith recommendations to provide guidance for health care professionals dealing with pregnant patients and those implemented in writing national health policies. We provide an international approach, but every country will need to adopt and follow their own country specific guidelines. The recommendations are written with the current available knowledge; this might change with evolving research.

The term “COVID-19” will be used to refer to both SARS-CoV-2, the corona-virus itself, and to the disease it causes. SARS-CoV-2 is a non-segmented, positive sense RNA virus. It is part of the family of coronaviruses (CoV) composed of four viruses that cause “common cold”, the Severe Acute Respiratory Syndrome (SARS) CoV and Middle East Respiratory syndrome (MERS) CoV. The latter two previously caused epidemics with high morbidity and mortality, especially in pregnant women [4,5,6,7,8,9]. COVID-19 is most closely related to SARS. It binds via the angiotensin-converting enzyme 2 (ACE2) receptor located on type II alveolar cells and intestinal epithelia [10].

The incubation time of COVID-19 virus is at a median of four days (interquartile range IQR of 2–7 days), with a range up to 14 days [11]. Generally, it causes a flu-like illness with constitutional signs and symptoms like myalgia and fever, and typical upper and, less frequently, lower respiratory symptoms (viral pneumonitis). The latter usually presents with cough, dyspnea and fever [12,13,14,15,16]. About 10% of cases present initially with gastrointestinal symptoms only (nausea, diarrhea) [17]. Anosmia has been reported to be an early symptom [18]. However, asymptomatic carriers have also been reported [11,19].

In the general population hospitalization occurs in about 23% of known COVID-19 positive cases (might be an overestimation since the amount of screening varies within each country) and among these the mortality rate is around 1%-2% (higher in Italy due to lack of resources and lower in Germany and South Korea, though depending on country specific reporting). Most fatalities are due to acute respiratory distress syndrome (ARDS) and multi-organ failure. High morbidity and mortality is seen in the elderly (>65 years of age) and patients with (multiple) comorbidities [13,16]. See Table 1 for an overview of high-risk profiles. Since the first cases were reported in December 2019 from an outbreak in the Hubei province in China, a worldwide pandemic emerged with a high disease burden and an increasing death toll, paralyzing the economy and pressuring social security networks as well as health care systems.

## 2. Susceptibility of Pregnant Women to COVID-19: Are Pregnant Women More Likely to Get Infected?

Although pregnant women are not immune-compromised in the classic sense, immunologic changes of pregnancy may induce a state of increased susceptibility to certain intracellular pathogens, especially viruses, intracellular bacteria and parasites [20,21]. Measles, primary varicella, influenza, variola (small pox), lassa fever, ebola and SARS are all examples of viral infections, where pregnant women are more susceptible to be infected and develop more severe complications of the disease and higher mortality compared to the non-pregnant population [4,5,20,22,23,24,25,26].

As for COVID-19, the reproductive rate or in other words the average number of people that an infected person transmits the virus to during the peak of the epidemic is between two and three (range 2.5–2.9), which is somewhat higher than seasonal influenza [27]. This number reflects both the virus’ characteristics and infection potential, as well as the human behavior (e.g., social distancing or not). The virus is transmitted by droplet-infection as well as surface-contact (face-to-fomite), with certain reported data describing persistence of viable virus on surfaces up to four days [11,28]. More evidence is needed on the possibility of airborne transmission and transmission during aerosol generating procedures and should be taken into consideration until further evidence is available [29,30,31]. Pre-symptomatic people can also transmit the disease, and for that reason, in China everyone is advised to wear face masks outside of the home environment [11,32].

Guideline 1. Every pregnant woman is considered high risk, as their susceptibility due to altered immune response may be higher, disease course is more severe and delivering intensive care is more difficult (see later). All pregnant women should take extensive preventative measures: hand hygiene and disinfecting surfaces with >60% ethanol and strictly adhere to measures of social distancing when interacting with other people [33]. This also accounts for their (household) partners.

Guideline 2. As for pregnant women working in high-risk-exposure settings (labor and delivery, operating theaters, respiratory wards, intensive care or high dependency units), transfer to low risk exposure settings is preferred. The exposure risk assessment should be done by every professional group individually and depending on the local endemic statistics.

## 3. Maternal Outcomes: Are Pregnant Women, once Infected with COVID-19, at Risk for Developing More Severe Disease?

Limited data is available on maternal outcomes in COVID-19 infection in pregnancy. However, in data from other viral illnesses such as influenza, SARS and MERS, pregnant women are more likely to develop viral pneumonitis, with higher morbidity and mortality compared to non-pregnant women [4,5,6,20,23,34,35,36]. Compared to non-pregnant women, these illnesses significantly increased maternal morbidity and mortality, especially in the second and third trimesters of gestation. Several explanations have been suggested, such as physiological alterations in cellular immune response during pregnancy and changes in pulmonary functions [20,21]. We would like to add to this the difficulty of managing severe pneumonia in pregnancy, such as intubation and mechanical ventilation, especially in the 3rd trimester where often (premature) delivery is imminent [37,38]. In addition, ventral positioning during ventilation, which is often required in severe COVID-19 cases, is not easy or feasible in late pregnancy.

Current data available on second and third trimester pregnancies with confirmed SARS-Cov-2/Covid-19 positivity (from a review of 32 cases in four case report series) have not reported maternal deaths [39,40,41,42]. One case, a 31 year old woman at gestational age (GA) of 34 weeks, presented with multi-organ failure and required mechanical intubation, ultimately leading to extracorporeal membranous oxygenation (ECMO) [40]. Another patient in Chen et al. series had concomitant pre-eclampsia at 36 weeks and was delivered by cesarean section without need for intensive care hospitalization.

Current data suggest lower morbidity and mortality for pregnant women with COVID-19 than during the SARS epidemic: for 6.3% (2/32) intensive care admissions versus 83% (5/6) and no mortality (0/32) versus 33% (2/6), respectively. The disease course tends to be rather mild and similar to non-pregnant women, generally presenting with flu-like constitutional symptoms (fever, fatigue, myalgia), cough and occasionally dyspnea [17,39,40]. Some pregnant patients present with laboratory abnormalities such as lymphopenia, thrombocytopenia and elevated liver enzymes [17,39,40]. Based on this limited data, we find pregnant women with COVID-19 to have similar rates of developing severe disease requiring intensive care (6.3%) as the general population (5%) [43].

There are no reports of maternal outcomes in first trimester pregnancies, but it may be too early since the start of the outbreak.

Guideline 3. With the limited evidence available at present, pregnant women seem to have a similar course of the disease compared to the general population. However, in previous outbreaks of similar respiratory viruses, pregnant women were not only more vulnerable but also had a more severe course of the disease. Critical care management of pregnant patients is more difficult (airway management, etc.). Therefore, until further data is available, pregnant patients above 24 weeks GA should be strictly protected from becoming infected and we advise pregnant women to be removed from high risk exposure workplaces such as certain health care workers (see above).

## 4. Pregnancy Complications: What Are the Risks for the Pregnancy?

Limited data is available on pregnancy outcomes in COVID-19 infection cases.

During the SARS epidemics an increased risk of miscarriage was reported during the first trimester in seven proven infected cases (4/7 miscarriages). When evaluating the data in detail, three of the four were as early as three to four weeks of gestation [4,35]. So far, for COVID-19 no first trimester cases have been published, but further research is awaited.

For second and third trimester outcomes we summarized the literature review of 31 singleton pregnancies at GA 25–39 weeks (Table 2) [39,40,41,42]. One third of COVID-19 positive patients (10/31) presented with preterm premature rupture of membranes (PPROM) and preterm labor and in 35.4% (11/32) fetal distress was reported. It is unclear how much of this is directly related to COVID-19 infection. Therefore, in proven cases of a new infection, alertness should be increased. Maternal hypoxemia can cause such complications. On the other hand, fever could also explain the increased risk of PPROM and preterm labor. One case of intrauterine fetal death is reported, in a case where the mother developed multi-organ failure requiring intensive care hospitalization and ECMO. When reviewing the data, we note a preterm birth rate of 53.6% (15/28). In COVID-19 positive patients a cesarean section rate of 96.4% was seen, possibly indicating that iatrogenic reasons (obstetrician’s fear) can also be a factor contributing to prematurity.

Intrauterine fetal growth restriction (IUGR) has so far not been reported in association with COVID-19 infection. Although IUGR is a known consequence of chronic maternal hypoxia, the effects of shorter and transient hypoxia in COVID-19 are unknown [44]. During the SARS-epidemic, small for gestational age neonates were reported in two women contracting the infection at 28 and 30 weeks GA and delivering at 33 and 37 weeks respectively [4]. In reported COVID-19 patients, delivery generally occurred within one week of diagnosis, making it impossible to assess the long term effect of transient maternal hypoxemia on fetal growth [39,40,41,42].

Guideline 4. Preterm delivery, PPROM and intrauterine fetal distress are potential complications of maternal COVID-19 infection, possibly caused by maternal hypoxemia. Further research is needed to confirm a causal relation. Cesarean section rates are vastly higher than in the general population, partly iatrogenic due to obstetricians’ insecurity.

Guideline 5. Timing of delivery should be determined by a multidisciplinary team, on a case by case basis, considering maternal and fetal clinical presentation. The rate of admissions of COVID-19 positive pregnant patients to an intensive care unit is similar to the general population (around 5%). Intensive care management after 24 gestational weeks is more difficult (airway management problems, fetal monitoring, etc.). Data is currently too limited to give definite general recommen-dations.

Guideline 6. Intrauterine growth restriction could be one potential long-term complication among patients recovering from COVID-19 infection, consistent with data obtained during the SARS epidemic. More data from COVID-19 patients is needed. Therefore, fetal growth should be followed up in COVID-19 infected pregnant patients. Additional ultrasound evaluation at gestational age 24-28-32-36 weeks with biometry, amniotic fluid measurement and assessment of uterine artery Doppler pulsatility index and midcerebral artery Doppler in case of IUGR <10th percentile) is indicated [45,46].

## 5. Fetal Risk: Does Vertical Transmission Occur?

When testing amniotic fluid, cord blood and neonatal throat swabs postpartum in six COVID-19 infected patients, Chen H et al. found no evidence of intrauterine vertical transmission [39]. Additionally, Liu et al. found no “serological” evidence of vertical transmission in 10 newborns. However, the method used for serological testing was not described [40]. Zhu et al. also described 10 neonates (8 singleton, 1 twin) with PCR-negative throat swabs for COVID 19 [42]. One case report based on positive IgM serology in a neonate suggested intrauterine infection [47]. The neonate was born by cesarean section to a COVID-19 positive mother with positive IgM and IgG antibodies (107.89 AU/mL and 279.72 AU/mL respectively; normal values IgM and IgG < 10 AU/mL). The neonate was isolated immediately from the mother and 2 h post birth, IgM and IgG titers were 45.83 AU/mL and 140.32 AU/mL respectively. However, five throat-swabs of the neonate for PCR from 2 h until 16 days post birth were performed to prove infection and were negative for COVID-19. Amniotic fluid, cord blood or placenta were not tested. Based on these findings, intrauterine infection seems unlikely and the IgG antibodies were most likely of maternal origin. The IgM result can be false positive, as seen before in cases of cytomegalovirus infection [48].

Pathology review of three placentas of confirmed COVID-19 positive patients following delivery by cesarean section showed no signs of villitis and chorio-amnionitis, and all three placental samples were negative for COVID-19 RNA [49].

Based on these reports, no evidence for intrauterine vertical transmission for COVID-19 in second or third trimester has been confirmed. These findings are in accordance with the findings regarding SARS-virus infections [50]. The expression of the ACE-2 receptor, necessary for the viral intracellular integration of COVID-19, seems to be weak in all cells of the fetal–maternal interface. This may explain the absence of maternal–fetal transmission across the placenta [10,51].

Guideline 7. Intrauterine vertical transmission of COVID-19 has so far not been reported, at least not between 25 and 39 gestational weeks. First trimester complications and teratogenic data do not exist. Based on the assumption that cells on the fetal–maternal interface are less susceptible to COVID-19 infection, the risk of first trimester complications is probably low. This message can be delivered to pregnant patients with COVID-19.

## 6. Mode of Delivery and Perinatal Transmission Risk: Is Vaginal Birth Safe?

Certain generalized viral infections, such as HIV, predispose to intrapartum neonatal transmission [39,52]. For COVID-19, data is limited. In one case series three neonates were born vaginally (one singleton, one set of twins) and throat swabs for PCR at day one of birth were negative for COVID-19 in all three cases [40]. Another COVID-19 positive patient had negative vaginal swab testing during delivery [53]. Thus, data suggest no increased risk of perinatal vertical infection transmission.

All indications for cesarean section in the nine cases reported above were maternal, i.e., fear of deterioration of COVID-19 pneumonia [39]. Only one case had an additional obstetrical indication for cesarean section (history of two previous cesarean sections) and two others had additional relative risk factors (one had pre-eclampsia and one had a history of two intrauterine fetal deaths). Two cesarean sections were performed for intrauterine fetal distress, suggested to be related to maternal hypoxemia. In another case report cesarean section was performed at GA 30 weeks due to a combination of maternal deterioration and fetal distress [41].

Of the reported neonates of COVID-19 positive mothers, half (4/8) were born preterm. Clinical outcomes in neonates reported seemed to be merely related to prematurity (mostly respiratory distress). One neonate died probably due to severe asphyxia following severe maternal COVID-19 disease with intensive care admission for multiple organ failure [42].

Guideline 8. Vertical transmission from passing through the birth canal is unlikely, but data are limited. Hence, if maternal condition is stable and proper fetal monitoring can be assured, vaginal delivery is preferred.

## 7. Neonatal Risks: Can Neonates Get COVID-19 after Birth? Is Breastfeeding Safe?

### 7.1. Postpartum Transmission to Neonates

Postpartum several reports confirm fresh newborns can become infected with COVID-19, the youngest 30 h postpartum [54]. A review describes three COVID-19 positive neonates with fever, cough, vomiting of milk and in two cases dyspnea but otherwise stable vital signs [55]. Another COVID-19 positive neonate showed stable vital signs, no fever or cough, but shortness of breath together with abnormal chest radiographs and abnormalities of liver function [56]. Considering that vertical intrauterine infection is unlikely, postpartum droplet or contact transmission from parents or other care takers to the neonate is the most plausible explanation [11,28,57]. Whether or not airborne transmission occurs is still not known [29,30].

Guideline 9. Postnatal transmission from parents or other care takers to the neonate is possible. Hence strict hygienic measures, including masks, hand hygiene and physical distancing (as far as possible) is recommended.

### 7.2. Breast Feeding

Breast milk samples of six COVID-19 positive mothers after giving birth showed negative PCR results [39]. Samples of breast milk of two women during the SARS-epidemic were also negative for the virus, although these women had most likely already recovered from the infection at time of birth [34,35]. Guidelines disagree, some indicate breastfeeding is allowed with precaution measures (RCOG). Alternatively, isolation of the neonate for 14 days is required in a separate neonatology ward until maternal illness resides in order to protect the neonate against postpartum vertical transmission (CDC and Chinese guidelines) [1,2,54,58,59].

Guideline 10. Vertical transmission via breast milk is unlikely. Neonates can be more at risk for developing (severe) complications of COVID-19 considering their immature immune system, based on limited data.

Two different approaches are proposed. A: the advantages of mother–child bonding and breastfeeding (with preventative measures such as wearing a surgical mask, hand-hygiene and washing the nipples before breastfeeding) outweigh the possible risk of neonatal infection (with the current limited data suggesting rather mild disease course in neonates) [2]. B: partially based on guidelines written after SARS and MERS-epidemics. The neonate is isolated on a neonatal ward for 10–14 days for surveillance and remains separated from the mother until clinical illness resides and precaution measures are lifted, preventing breastfeeding. Since transmission through breast milk seems unlikely (see above), pumping milk and bottle-feeding can be considered [1,58,60]. Until further evidence, parents should be counseled about the risks and benefits of both approaches.

## 8. Diagnostics: How to Diagnose COVID-19 in Pregnancy? Can/Should a CT-Scan Be Performed?

There appears no reason that diagnosis of COVID-19 in pregnancy should be different from the general population. The threshold for testing pregnant patients presenting with suspicious symptoms should be low, considering the possibility to provide more close follow-up for fetal and maternal complications. Hypoxia does not correlate with auscultation nor with chest imaging. Therefore, pulse oxygen saturation, tachypnea and dyspnea are important clinical signs. However, note that also silent hypoxia (hypoxia presenting without respiratory distress or dyspnea) has been reported.

### 8.1. Polymerase Chain Reaction (PCR) Testing

Ideally, the diagnosis of COVID-19 infection is made by performing nasopharyngeal swabs for PCR testing in people presenting with suspicious symptoms. It is important to note, that while the specificity of this PCR is near 100% (few false positive cases), sensitivity is rather low between 66% and 80% [61]. This could partly be due to sampling error, as deep intranasal and throat swabs are required but are not always easy to obtain. It is estimated that people with more severe disease may have higher viral load. Likewise, sampling early in the disease course may have lower sensitivity than sampling later. Therefore, if the PCR test is negative but suspicion for COVID-19 remains, ongoing isolation and resampling 24 h to several days later is recommended. In general, two subsequent negative samples rule out the infection. There is no difference in swabbing or test performance between pregnant and non-pregnant women.

Guideline 11. The final diagnosis of COVID-19 infection in pregnant women is a deep nasal and/or throat swab for PCR. As sensitivity of PCR test is estimated between 66% and 88% and thus around one out of four are false negative, in a suspicious case who tested negative, continued precaution measures should stay in place and a repeat swab after (minimum) 24 h is recommended. As both maternal and fetal complications can occur, we advise a low threshold for testing pregnant women.

### 8.2. Computed Tomography (CT) Scanning

There have been reports of high sensitivities when conducting CT scanning of the thorax for early diagnosis of COVID-19, even surpassing the PCR tests [62]. In one study CT scanning was performed in 15 healthcare workers who were exposed to COVID-19 before they became symptomatic. Ground glass opacification on CT scan was seen in 14/15 patients. Guan et al. and AIi et al. found that in patients with a confirmed positive COVID-19 PCR, CT-scan was positive in 840/975 cases and 580/601 cases and thus resulting in sensitivities of 86% and 97%, respectively [14,61]. However, several problems arise: (1) precise definition of what constitutes a “positive” CT scan is currently still lacking. (2) Findings are nonspecific, as any viral pneumonia can show similar findings on CT. (3) Among patients with constitutional symptoms only (fever, myalgia, malaise, etc. but without respiratory symptoms), a CT scan may be less sensitive (around 50%) [63]. (4) Although radiation exposure of CT scanning is low enough not to be harmful for the fetus, especially when protected by a lead protection covering the abdomen, the exposure of breast tissue to radiation can still be harmful, increasing breast cancer risk in the future [64,65]. A single chest x-ray, mammogram and CT-scan exposes the patient to about 0.1 milli-sievert (mSv), 0.4 mSv and 10 mSv, respectively. This corresponds to the natural background radiation exposure of 10 days, 7 weeks or more than 3 years respectively [66]. When using low dose chest CT, the mean effective radiation dose is 1.4 mSv (standard deviation = 0.5) and is thus estimated acceptable [67]. Several centers are now applying routine low dose chest CT scanning for COVID-19 screening of all patients being admitted. This allows detection of COVID-19 positive patients with high sensitivity (including asymptomatic) and allows early triage.

Guideline 12. Diagnosis of COVID-19 patients is similar to non-pregnant patients. PCR testing of nasopharyngeal swabs is routine. However, due to low sensitivity of this test, repeat testing after a minimum of 24 h is advised in the case of a negative result with clinical findings consistent with COVID-19. Low dose CT-scanning can also be used as a screening test. Dosages are relatively safe during pregnancy when abdominal shielding is used. The risks of radiation on the breast tissue are to be balanced against the benefits of higher sensitivity and possibility of early triage of COVID-19 positive patients upon admission.

### 8.3. Serologic Testing

Currently no commercial antibody tests are available and approved, and several laboratories are working to develop reliable serologic tests. Preliminary results of newly developed ELISA-tests and humoral response have been reported in China and the United States but further research is needed [68,69]. Using an ELISA-assay in suspected COVID-19 cases with initial negative qPCR, IgM and IgA antibodies were detected after a median of 5 days (IQR 3-6), and IgG was detected after 14 days (IQR 10-18). The detection efficiency by IgM ELISA is higher than that of the qPCR method after 5.5 days of symptom onset. Combining IgM ELISA assay with PCR has a higher sensitivity (98.6%) compared to qPCR test alone (51.9%) [68].

As it is currently unknown how long antibodies against COVID-19 last after primary infection, repetitive antibody testing will be crucial to assess long term immunity in order to develop future vaccines. In addition, several COVID-19 strains with different virulence have been reported [70,71]. It is not yet known how fast the virus mutates, creating strains for which previously infected (or vaccinated) individuals would no longer be immune to. It will be important to include pregnant woman in vaccination trials, since they are considered a high-risk population [50].

Guideline 13. As soon as serologic tests are available, pregnant women should be tested primarily. For assessing vaccine efficacy maternal vaccination should be considered early in the design of these trials.

## 9. Hospitalization: When to Hospitalize a Pregnant Patient with Suspected or Confirmed COVID-19?

All pregnant patients with suspected symptoms should undergo testing. Clinical assessment will determine the need for hospitalization while awaiting the test results.

We proposed the following criteria for hospitalization in pregnant women based on the Modified Early Obstetric Warning Score (MEOWS, Table 3), and proposed care for COVID-19 pregnant patients by Peyronnet et al. and Liang H et al. [9,72,73,74,75].

### 9.1. No Hospitalization

Pregnant with mild disease and no comorbidities (see Table 1 above): symptomatic but absence of dyspnea and stable vital signs.

The patient will be asked to follow up clinical parameters daily at home: fever, respiratory rate, blood pressure and fetal movements.

Contact by phone with obstetric health care provider every 48 h for reporting signs and symptoms or earlier in case of subjective deterioration or abnormal parameters based on MEOWS reference values (Addendum 1).

### 9.2. Hospitalization (Referral to Tertiary Centre Depending on Gestational Age and Local Policy)

Pregnant with moderate disease or pregnant with mild disease and comorbidities (Table 1)

(A) Acute community acquired pneumonitis with oxygen requirement*:

-desaturation < 96% O2 on ambient air.

-or tachypnea ≥ 21 respirations/minute on ambient air.

-or clinical evident signs of dyspnea.

OR

(B) Signs of lower respiratory infection without oxygen requirement but with comorbidities (Table 1)

Isolation and infection prevention measures to be taken as described below.

### 9.3. Hospitalization on Intensive Care Unit with Consulting Obstetric Support (Refer to Tertiary Center Depending on Gestational Age and Local Policy)

(A) Pregnant with severe disease: respiratory rate ≥ 30/min, resting SaO_2_ < 94%, arterial blood oxygen partial pressure (PaO_2_)/oxygen concentration (FiO_2_) ≤ 300 mmHg.

OR

(B) Pregnant with oxygen requirement (see above) and comorbidities. (Table 1)

OR

(C) Pregnant with critical disease: shock with organ failure, respiratory failure requiring mechanical ventilation or refractory hypoxemia requiring ECMO.

To be determined and managed on a case by case basis by a multidisciplinary team (senior obstetrician, internal medicine specialist/pulmonologist, intensive care specialist, infectious disease specialist).

Guideline 14. Pregnant patients need to be assessed according to their respiratory symptoms severity score and potential presence of comorbidities. In severe/critical cases, immediate referral to a tertiary center is indicated.

## 10. Treatment Options: Which Medications for COVID-19 Infection Are Safe to Use When Attempting Treatment in Pregnant Patients?

### 10.1. Corticoids

Clinical evidence does not support the use of corticosteroid treatment for COVID-19 related lung injury [76]. However, short term administration of corticoids (either betamethasone or dexamethasone as per local protocol) intramuscularly to improve fetal lung maturity when preterm delivery is imminent should be considered and is not harmful [9].

### 10.2. Antivirals

Application of antiviral treatment should be implemented by per region specific protocols. All protocols are experimental. Current knowledge about antiviral treatment in pregnancy is summarized below and based on the Belgian interim guidelines consensus paper [77]:

#### 10.2.1. Chloroquine and Hydroxychloroquine

Chloroquine has good in vitro activity against COVID-19 and seems to reduce the duration of viral shedding. This does not necessarily translate to clinical efficacy (many previous experiences were disappointing). Clinical trials are ongoing. The therapeutic window is quite narrow (cardiotoxicity/arrhythmia), requiring caution with higher dosages.

Hydroxychloroquine may be more effective than chloroquine in vitro, so that lower dosages could be used [78]. Results of Gautret’s study, very recently released, confirm that viral positivity in respiratory secretions (measured by PCR) is significantly decreased at day 6 in hydroxychloroquine-treated COVID-19 patients (*n* = 26) versus those with supportive care (*n* = 16 controls): 30% positivity versus 87.5%, *p* < 0.001) [79]. However, the study has several limitations, acknowledged by the authors. These preliminary results support the current choice of hydroxychloroquine as first-line treatment. Benefit of additional azithromycin in the study remains controversial [77,79].

General precautions of hydroxychloroquine and chloroquine are lengthening of QTc-interval and known drug interactions (check at http://www.covid19-druginteractions.org). It is contraindicated in patients with myasthenia gravis, porphyria, retinal pathology and epilepsy.

Chloroquine has been used for decades (at a total of 25 mg/kg within three days) for malaria treatment without any monitoring and side effects, including in pregnant women. However, exposure to high dosages is limited [80]. Long-term daily use of hydroxychloroquine in pregnancy is not teratogenic. However, this conclusion is based on small case series (low level evidence) [80].

Dosage: Chloroquine base 600 mg (10 mg/kg) at diagnosis and 300 mg (5 mg/kg) 12 h later, followed by 300 mg (5 mg/kg) twice daily up to day 5, OR Chloroquine phosphate 1000 mg at diagnosis and 500 mg 12 h later, followed by 300 mg twice daily up to day 5.

Hydroxychloroquine 400 mg at diagnosis and 400 mg 12 h later, followed by 200 mg twice daily up to day 5.

#### 10.2.2. Lopinavir/Ritonavir

Lopinavir/ritonavir is an antiretroviral protease inhibitor used in the treatment of HIV. It does not provide clinical benefit in hospitalized patients with COVID-19 and has no impact on viral shedding. This is in line with in vitro experiments with SARS-CoV2 but also SARS-CoV1. Lopinavir/ritonavir can still be considered a second choice when hydroxychloroquine is contraindicated.

Lopinavir/ritonavir are known to have potential severe side effects like pancreatitis, arrhythmia, severe allergic reactions, hepatotoxicity and drug interactions.

Lopinavir/ritonavir associated regimens for HIV treatment in pregnancy seem to have a higher rate of adverse birth outcomes. After adjustment for maternal age, gravida, and educational attainment, singleton infants exposed to tenofovir disoproxil-emtricitabine-efavirenz (TDF-FTC-EFV) from conception were less likely to have (severe) adverse birth outcome when compared with infants exposed to tenofovir disoproxil-emtricitabine-lopinavir/ritonavir (TDF-FTC–LPV-R) (Adjusted Relative Risk ARR, 1.31; 95% CI, 1.13–1.52). The lopinavir/ritonavir group showed a significantly higher rate of being small for gestational age (<10^th^ percentile) neonates (ARR, 1.56; 95% CI, 1.25–1.97) [81]. Due to low evidence of efficacy in treating COVID-19 related complications and the increased risk for adverse birth outcomes, the use of lopinavir/ritonavir in pregnancy is not advised. If considered, the advice of an infectious disease and obstetric specialist is needed.

Dosage: lopinavir/ritonavir 400/100 mg (= 2 tablets of 200/50 mg) twice daily for 14 days.

#### 10.2.3. Remdesivir

Currently used only in clinical trials, i.e., safety profile and efficacy need to be further determined. No data is available in pregnancy.

Guideline 15. As no treatment for COVID-19 is established yet, all drug trials must be considered experimental, and this message must be given to patients. Corticosteroids only have a place in prevention of neonatal lung hypoplasia, necrotic enterocolitis and interventricular hemorrhage due to prematurity. (Hydro)chlorquinine has a reasonable safety profile in pregnancy but general precautions must be taken into consideration. Antiretrovirals are currently tested in clinical trials but have no place in pregnancy, unless no other treatment options are available to safe maternal life.

## 11. Organization of Health Care Facility: How to Organize in- and out-Patient Clinics? Which Isolation Precautions to Take? What Protective Equipment Is Needed for Health Care Personnel?

### 11.1. Ambulatory Obstetric Care

Guideline 16. During an epidemic, routine obstetric follow-up consultations should be limited to the strict minimum in order to minimize exposure risk for both patients and health care providers (social and physical distancing). We summarized our recommendations in Table 4.

Obstetric patients presenting with alarming symptoms for emergency consultation should contact their obstetric care provider by telephone to determine whether assessment in the hospital is necessary.

Guideline 17. All pregnant patients contacting a health care provider with symptoms suspected for COVID-19, should be directed by telephone to a specific COVID-19 triage unit (as per regional protocol) for further evaluation and testing for COVID-19. Depending on clinical presentation the patient will be either hospitalized with isolation measures or transferred to home isolation, results pending.

### 11.2. Hygiene Preventative Measures

International guidelines (CDC, ANZICS, WHO, RCOG) disagree on preventative measures to prevent airborne infection, whether or not to use FFP2/N95 ultra-filtration masks for health care providers in all contact with (possible) COVID-19 positive patients or only in aerosol producing procedures [2,3,28,29,30,31,57,82,83]. In our opinion, health care providers working in close contact with COVID-19 suspected or positive patients should always wear full personal protective equipment (PPE) independent of performing “aerosol producing” procedures (waterproof gown, gloves, glasses or face shield) with FFP2 or N95 masks instead of surgical masks. Examples for close contact are daily nutritional assistance, bathing, surgical procedures, in labor and delivery wards or on wards providing airway care. Implementation depends on country-specific guidelines and availability of FFP2/N95 masks.

Guideline 18. All health care providers should apply extensive hand hygiene and surface hygiene protocols, wear gloves when in contact with patients or medical material and consider wearing a surgical mask to limit infection transmission to the patient, especially if coughing (health care personnel can be asymptomatic COVID-19 carriers). In the case of any potential signs of illness related to COVID-19, even if minor, the caregiver should stay home until the symptoms disappear and tests negative or for a minimum of 14 days if not tested.

Guideline 19. People working in close contact with COVID-19 suspected or proven positive patients should always wear full personal protective equipment (waterproof gown, gloves, glasses or face shield and FFP2 or N95 masks). Implementation depends on country-specific guidelines and availability of FFP2/N95 masks. This includes all (health) care providers and spans a wider extent of procedures beyond officially recognized “aerosol” producing procedures: all nutritional and bathing care, all surgical procedures, during labor and especially during delivery or when performing airway manipulation or care. In all these circumstances airborne transmission could be more likely.

### 11.3. Assessing Obstetric Emergency in a Potential COVID-19 Infected Patient

Guideline 20. All patients should be asked to call the health care facility prior to visit. The obstetric health care provider should determine the level of “obstetric emergency” (as by clinical expertise) and ask about symptoms and possible risk contacts (close contact with people presenting with COVID-19 related symptoms or tested positive in the last 14 days). This triage determines the level of precautions taken during hospitalization or delivery (Table 5, Table 6 and Table 7).

### 11.4. Management of COVID-19: Hospitalization on Obstetric Ward

See Table 6.

### 11.5. Labor and Delivery Ward.

See Table 7.

## 12. Care in Labor of COVID-19 Positive Women

### 12.1. Initial Assessment

For all patients with suspicion of COVID-19 upon admission on the labor and delivery ward, screening tests for COVID-19 are indicated as per local protocol (PCR swab and/or CT scan, maybe serology in the future). Clinical maternal parameters need to be assessed hourly, based on the MEOWS score system (Table 3). It is imperative to have monitoring of peripheral oxygen saturation (aim to keep O2 sat >94%) and respiratory rate (<20/min), fever (aim to keep temperature <38.5 °C) and blood pressure. Healthy pregnant women tend to compensate long with normal oxygen saturations, thus respiratory rate should be monitored closely.

Even if the COVID-19 infection seems to be the most important finding, care must be taken to rule out underlying pathologies, such as pre-eclampsia, cardiac pathology, pulmonary embolism, etc. Screening for coinfection with influenza is also recommended. If indicated, other infections such as respiratory syncytial virus, mycoplasma, *Streptococcus pneumonia* and legionella should be tested for. Bacterial blood cultures should be performed in patients presenting with lower respiratory symptoms and fever.

Complete blood count with differential, kidney function with electrolytes including calcium and magnesium levels, liver function tests (lactate dehydrogenase) and coagulation tests (INR, PTT, fibrinogen), C-reactive protein and procalcitonin, NT-proBNP, troponin should be done at admission [84]. D-dimers are generally elevated in pregnancy and thus not reliable [85]. Lymphopenia is common in COVID-19, but when presenting with increased neutrophil count, bacterial sur-infection is likely and should be treated accordingly.

Arterial blood gas should be performed in severe cases presenting with desaturation <94%. Pregnancy related adaptations (respiratory alkalosis with a normal p_a_CO2 of 28–32 mmHg) should be taken into account when interpreting as shown in Table 8 [86,87].

Guideline 21. Assessment and continued follow-up assessment of respiratory function and general condition of laboring patients with COVID-19 infection are important, with caution to equally consider other underlying pathologies or alternative diagnoses. It is advised to account for pregnancy adaptations when oxygen saturation is decreased, in consultation with your pulmonologist.

### 12.2. Management during Labor

#### 12.2.1. General Supportive Measures

Position the patient in the left lateral tilt or upright positions to minimize vena cava compression. Oxygen should be provided with nasal cannula or face mask for maternal indications only, as it shows no intrapartum fetal benefit [88]. Fluid restriction is advised especially in oxygen dependent patients, avoid fluid boluses and even maintenance infusion [3,31]. Close monitoring of fluid balance is advised and should be close to zero.

#### 12.2.2. Antibiotic Prophylaxis

-GBS-prophylaxis with penicillin G or ampicillin as per local protocol.

-Additionally, prophylaxis for bacterial secondary infection in case of COVID-19 pneumonitis, generally ceftriaxone 2g I.V. once daily during 5–7 days, is advised. Benefit of additional azithromycin (covering atypical bacteria) was proposed improving outcomes in patients using hydroxychloroquine as antiviral treatment, but results are controversial (azithromycin 500 mg loading dose, then 250 mg once daily for four days) [77,79,84].

#### 12.2.3. Antiviral Treatment

To be applied based on local protocol. In case (hydroxy)chloroquine is used, toxicity should be closely monitored with daily electrocardiogram (QTc-prolongation), glucose monitoring every 4 h and daily laboratory tests (complete blood count, liver and kidney function, electrolytes) [77,84].

Guideline 22. Symptomatic COVID-19 infected patients are considered high risk, especially when pregnant. Oxygen therapy is indicated to keep oxygen saturation ≥94%, fluid restriction with a fluid balance close to zero is advised, antibiotic prophylaxis should be applied for preventing secondary infection in case of COVID-19 pneumonia, based on local protocol. Antiviral treatment (if applied, preferably hydroxychloroquine) can be given and should be based on local protocol. Close monitoring in that case is warranted.

#### 12.2.4. Obstetric Medications and Safety Profiles

See Table 9 [89,90,91,92,93,94,95,96].

Guideline 23. When using specific obstetric medication in COVID-19 patients, specific interactions may occur. This requires special caution and close observation. Recommendations are summarized in Table 9.

### 12.3. Induction of Labor

Inductions of labor for medical indications should not be postponed. In case of indication for induction of labor, all COVID-19 suspected patients should be screened according to their level of emergency upon admission, as discussed in the former chapter. If the patient is stable, she will be asked to go home and return for induction when test results are known. If obstetric emergency: admit to ward in isolation and treat as possible COVID-19 positive until screening results are known.

If COVID-19 is diagnosed early in the disease course in a term pregnancy, induction of labor is indicated to avoid complications, considering the severity of the disease generally peaks in the second week [95]. Maternal condition is primary to fetal condition, and in case of deteriorating maternal condition an expedited delivery is urged. It should not be delayed in order to complete fetal lung maturation. Where possible vaginal birth should be attempted.

Prostaglandins and Foley catheters can be used according to the local protocol [95]. Oxytocin, however, has to be used with care, as it is associated with an increased risk of fluid overload when administered in bolus or in high dosages, which can worsen critical cases of COVID-19 [89,90,91,93].

Guideline 24. Induction of labor in suspected COVID-19 positive cases for medical or obstetric indications should not be postponed. Depending on the urgency, COVID-19 test results should preferably be obtained before admitting the patient to the hospital. Induction methods can be used as per protocol, but extreme care is warranted in not using overly high dosages or boluses of oxytocin due to the risk of fluid overload and cardiovascular decompensation in critically ill patients.

### 12.4. Delivery Care

Limit the number of people in the room.

Everybody should be trained in donning and doffing PPE (see above) [29,30,82].

Every “isolation” delivery room should have a basic provision of the following equipment that stays in the room until after delivery: a delivery set, CTG or Doppler monitor, a monitor for maternal vital signs including saturation, material for vacuum extraction and/or forceps, suturing material, provision for intravenous access and fluid administration, urinary single use and Foley catheters, oxygen masks, nasal cannula, a Bakri-balloon (depending on availability and local protocol), a stethoscope and an adult ventilation balloon with mask. It should also contain a basic medication set containing oxytocin, prostaglandins (when used in treatment of postpartum hemorrhage), tranexamic acid, penicillin or ampicillin, ceftriaxone, magnesium sulfate bolus dose, glucose 50%, lidocaine or related medications for local anesthesia, epinephrine and crystalloid fluids.

Neonatologist should be present at birth since neonatal complications can occur. The neonatology team should be alerted sufficiently ahead of time to allow time for donning of PPE.

Guideline 25. Limit the people at delivery as much as possible and make sure everyone present is trained in donning and doffing of PPE. A (neonatology) team member trained in newborn resuscitation should be present in case neonatal complications occur and should be alerted sufficiently ahead of time to allow time for donning of PPE.

### 12.5. Third Stage of Labor and Postpartum

Prostaglandins are safe to use in COVID-19 patients, but further evidence is needed to confirm.

Oxytocin can be used with care in COVID-19 patients (see above). A bolus of five international units (IU) at time of delivery of the first shoulder for active management of labor can be safely used. A second slow bolus of 10IU oxytocin can be applied and continuous oxytocin infusion of 10 IU/h until a maximum of 60IU/24 h in case of uterine atony should also be safe to use.

Methylergometrine should not be used in COVID-19 patients, since cases of acute respiratory failure following the administration have been reported [92,94].

Tranexamic acid is safe in COVID-19 patients, based on medication characteristics. Further evidence is needed to confirm.

Delayed cord clamping is not advised until more research confirms the finding of absent vertical transmission.

A separate “isolation room” for the evaluation and resuscitation of neonates born to COVID-19 positive mothers should be installed. Depending on local policy the neonate will be isolated in the neonatology ward or remain with the mother after birth (as discussed earlier).

Deep venous thrombosis prophylaxis should proceed like it would be decided otherwise. But also, all parturients diagnosed with COVID-19 should receive LMH for at least 10 days, even in the absence of other risk factors It should even be considered to increase the dose of the low molecular weight heparines (LMWH) in severely ill patients, as according to one study about 27% of patients had venous thromboembolic events and 4% had arterial thromboembolic events despite normal prophylactic doses of LMWH [97]. Consequently, these authors suggested doubling the typical dose of prophylactic heparin (e.g. enoxaparin 40 mg twice daily, rather than once daily).

Guideline 26. During postpartum, be careful with oxytocin and do not use methylergomethrine for management of uterine atony. Evaluate the neonate immediately after birth in a COVID-19 designed isolated resuscitation room. Depending on local protocol, the neonate will remain in isolation in the neonatology ward or will be reunited with the mother. LMWH for thrombosis prophylaxis should be performed, and its dose preferably doubled in case of severe COVID-19 illness. 

### 12.6. Analgesia in Labor

#### 12.6.1. Peridural/Spinal (Neuraxial) Anesthesia

-Neuraxial anesthesia is not contraindicated in COVID-19 [2].

-The possibility of thrombocytopenia due to COVID-19 should be ruled out.

-Decide in time for neuraxial anesthesia to minimize (the more frequent) need for general anesthesia in the event of emergency cesarean section.

#### 12.6.2. Inhalation Sedatives (Nitric Oxide, etc.)

Inhalation sedatives should be avoided in COVID-19 since it increases the risk of forming aerosols, potentially increasing the risk of exposure for health care providers.

#### 12.6.3. Opioid Pump Analgesia

Should be avoided in COVID-19 since respiratory suppression can occur.

#### 12.6.4. General Anesthesia

General anesthesia should be avoided when possible since intubation increases the risk of creating aerosols, potentially increasing the risk of exposure for health care providers.

Guideline 27. Peridural/spinal analgesia is preferred. Inhalation or general anesthesia should be avoided.

## 13. Partners of a Suspected or Confirmed COVID-19 Positive Woman: Can a Partner Be Present at Birth? Can a Partner Visit the Patient?

(Household) partners of patients presenting with symptoms, with a possible risk contact or confirmed COVID-19 positive will be considered and treated as COVID-19 positive.

Guideline 28. A partner of a COVID-19 positive woman must be considered COVID-19 positive. Whether a partner can be present at birth should depend on local policy and availability of personal protective equipment. It is imperative to minimize the risk exposure of the health care personnel present.

## Figures and Tables

**Table 1 diagnostics-10-00243-t001:** Criteria for people at high risk for severe illness with COVID-19.

➢ **People Aged 65 Years and Older** ➢ **People who live in a nursing home or long-term care facility**	
Other high-risk conditions could include:	
➢ **People with underlying end organ dysfunction**	
Chronic lung disease (mucoviscidosis, chronic obstructive lung disease, moderate to severe asthma or any other lung disease that could deteriorate with viral infection)	
Serious heart conditions (New York Heart Association classification NYHA 3-4, heart valve disease, history of cardiac surgery or coronary artery disease)	
Severe renal insufficiency (requiring hemodialysis)	
Severe hepatic disease (liver cirrhosis ≥Stage 4)	
Diabetes mellitus (poorly controlled insulin-dependent or with complications such as micro-and macro-angiopathy)	
Severe obesity (body mass index [BMI] >40)	
Metastasized cancer	
➢ **People who are immunocompromised**	
Drug-induced (chronic steroid use or other agents that suppress immunity)	
Organ transplant patients under immunosuppression	
Hematological malignancies	
Cancer therapies (chemotherapy etc.)	
Poorly controlled HIV-infected with CD4 < 200/mm	
➢ **People who are pregnant**	
Based on CDC COVID-19 guidelines, Belgian COVID-19 guidelines, CNGOF COVID-19 and pregnancy guidelines.	© ISIDOG COVID-19 2020 guidelines

**Table 2 diagnostics-10-00243-t002:** Second and third trimester singleton pregnancy outcomes in 31 confirmed COVID-19 positive patients.

Clinical Pregnancy Outcome	Chen H et al. (Lancet)	Liu et al. (J. of Infection, Prepress)	Zhu et al. (Transl. Pediatr.)	Wang X et al. (Clin. Infect. Dis.)	Total %
	*n* = 9	*n* = 13	*n* = 8	*n* = 1	
Median maternal age (years; range)	28; 24–40	30; 22–36	30; 25–35	30	
Median gestational age at diagnosis (weeks; range)	37; 36–39	35; 25–39	35; 33–39	30	
Intensive care hospitalization	0/9	1/13	0/8	1/1	6.3%
Mechanical ventilation	0/9	1/13	0/8	0/1	3.2%
Maternal mortality	0/9	0/12 *	0/8	0/1	0%
Delivery within 1 week after diagnosis	9/9	NA **	8/8	1/1	100% *
Intrauterine fetal distress during hospitalization	2/9	3/13	5/8	1/1	35.4%
PPROM/preterm labor	1/9	7/13	2/8	0/1	32.3%
Premature delivery (<37 weeks)	4/9	6/10 **	4/8	1/1	53.6%
Extreme premature delivery (<34 weeks)	0/9	NA **	1/8	1/1	11.1%
Mors in utero	0/9	1/13	0/8	0/1	3.2%
Neonatal vertical transmission	0/6 ***	0/10 **	0/7 ***	0/1	0%
Cesarean section	9/9	10/10 **	7/8	1/1	96.4%

* One patient was on extracorporeal membranous oxygenation at the time of publication, the outcome is unknown. ** Gestational age at the time of delivery was not reported, 3 patients were discharged home after clinical remission, delivery data on these patients is lacking. Thus 100% delivery within 1 week of infection is an overestimation. *** Data of throat swabs in 6/9 neonates Chen et al. and 7/8 Zhu et al.

**Table 3 diagnostics-10-00243-t003:** Modified early obstetric warning score (MEOWS).

MEOW Score	3	2	1	0	1	2	3
O2 saturation (%)	≤85	86–89	90–95	≥96			
Respiratory Rate (breaths/min)		<10		10–14	15–20	21–29	≥30
Heart Rate (beats/minute)		<40	41–50	51–100	101–110	110–129	≥130
Systolic blood pressure (mmHg)	≤70	71–80	81–100	101–139	140–149	150–159	≥160
Diastolic blood pressure (mmHg)			≤49	50–89	90–99	100–109	≥110
Diuresis (mL/h)	0	≤20	≤35	35–200	≥200		
Central nervous system level			Agitated	Alert/awake	Response only to verbal stimuli	Response only to pain stimuli	No response
Temperature (°C)		≤35	35–36	36–37.4	37.5–38.4	≥38.5	
**MEOWS 0–1**	Normal						
**MEOWS 2–3**	Abnormal but stable, report findings to health care provider the same day.						
**MEOWS 4–5**	Abnormal and unstable, to be evaluated by medical doctor within 30 min.						
**MEOWS ≥ 6**	Abnormal and critical, to be evaluated by medical doctor within 10 min.						
© ISIDOG COVID–19 2020 Guidelines							

**Table 4 diagnostics-10-00243-t004:** Proposed follow-up schedule for pregnant patients in COVID-19 epidemic.

Gestational Age		Advised Follow-Up Plan
**11–13+6d**	**Weeks**	intake consultation documentation and risk stratification + blood type, complete blood count and serological testing (as per standard protocol) + clinical parameters + structural ultrasound scan (+/− trisomy screening)
**20–22**	**Weeks**	clinical parameters + structural ultrasound scan + arranging appointment for glucose challenge test if indicated (as per country specific protocol) + instructions for alarm symptoms + at home follow up of blood pressure (BP) at 24 and 28 weeks
**24–28**	**weeks**	glucose challenge test ambulatory (without consultation) as per country specific protocol (general screening or risk selection)
**30–32**	**weeks**	clinical parameters + fetal biometry ultrasound scan + instructions for alarm symptoms + at home follow up of BP 2-weekly
**34–36**	**weeks**	clinical parameters + Group B streptococcal sampling (as per country specific protocol) + delivery planning + instructions for alarm symptoms + at home follow up of BP weekly
If a pregnant patient is positive for COVID-19 - routine consultations should be postponed by 14 days. If a pregnant patient is assessed high risk and needs additional follow-up this needs to be assessed case by case. We advise partners to be absent for routine consultations, to limit the exposure risk for health care providers. © ISIDOG COVID-19 2020 Guidelines		

**Table 5 diagnostics-10-00243-t005:** Triage and outpatient action plan according to obstetric risk and COVID-19 infection status.

Action Plan According to Obstetric Risk and Covid-19 Infection Status		
Obstetric Emergency	LOW	HIGH
**COVID-19 symptoms**	COVID-19 triage unit + testing	Admission in special isolated room in obstetrical ward
	Postpone obstetric visit until test result is known	All isolation and protection measures in place (see below)
**Contact COVID-19 only**	Outpatient visit possible	Restricted visit
	Patient wears mask and gloves	
**No contact/no symptoms**	Outpatient visit possible	Normal obstetric ward admission
	Hand hygiene + social distance	
© ISIDOG COVID-19 2020 Guidelines		

**Table 6 diagnostics-10-00243-t006:** Management of hospitalization on obstetric ward in COVID-19 endemic.

**NO COVID-19 Symptoms or contact**	Normal room on obstetric ward.
	Instruction to patient of hygienic measures.
	Health care workers: hand hygiene, gloves, surgical mask.
**Contact COVID-19 and/or Symptoms**	Designated isolation room in obstetric ward (negative pressure if available and at distance from other obstetric patient’s rooms).
	All personnel entering the room wearing full personal protective equipment (PPE): waterproof gown, goggles of eye-shield, surgical mask, gloves.
	If symptomatic, patient wears surgical mask + hand hygiene.
	Cardiotocographic (CTG) monitor and medical material should not leave the patient’s room.
	Limited personnel, PPE trained, who do not give care to other pregnant patients.
	No visitors.
	All precaution isolation and infection prevention measures stay in place until COVID-19 test result is known. © ISIDOG COVID-19 2020 Guidelines

**Table 7 diagnostics-10-00243-t007:** Management of hospitalization on obstetric ward in COVID-19 endemic.

All Patients upon Admission.	All Referred to a Designated Triage “Isolation Room” on the Labor and Delivery Ward Where Risk Stratification Should Be Done.
**NO COVID-19 symptoms or contact**	Normal labor/delivery room.
	Instructions to patient of hygienic measures.
	Health care workers: hand hygiene, surface hygiene, gloves, surgical mask.
**Contact COVID-19 and/or symptoms**	Designated isolation room in obstetric ward (negative pressure if available and at distance from other obstetric patient’s rooms).
	All personnel entering the room wearing full personal protective equipment (PPE): waterproof gown, goggles of eye-shield, surgical mask, gloves.
	If symptoms, patient wears surgical mask + hand hygiene
	Cardiotocographic (CTG) monitor and medical material should not leave the patient’s room.
	Health care providers present at delivery wear full PPE, with FFP2 or N95 mask (depending on availability).
	Same precautions for cesarean section, whether or not general anesthesia is applied.
	No “gentle cesarean section”, as this requires extra personnel and complicates social distancing and extra use of PPE.
	Limited personnel, PPE trained, who do not give care to other pregnant patients
	One partner at birth optionally (see below)
	All precaution isolation and infection prevention measures stay in place until COVID-19 test result is known.
© ISIDOG COVID-19 2020 Guidelines	

**Table 8 diagnostics-10-00243-t008:** Normal values for arterial blood gases in pregnant and non-pregnant women.

Arterial Blood Gas Characteristic	Normal Values	Values in Pregnancy	
pH	7.34–7.44	7.40–7.46	Increased
Arterial oxygen partial pressure (PaO2)	10–13 kPa		Unchanged
	75–100 mmHg		
Arterial carbon dioxide partial pressure (PaCO2)	4.7–6.0 kPa	3.7–4.2 kPa	Decreased
	35–45 mmHg	28–32 mmHg	
HCO3- (bicarbonate)	22–26 mEq/L		
SBCe (standardized bicarbonate)	21–27 mmol/L	18–21 mmol/L	Decreased
Base excess	−2 to +2 mmol/L		Unchanged
*kPa = kilopascal, mmHg = millimeters of mercury, mmol = millimol, L = liter, mEq = milliequivalent*			

**Table 9 diagnostics-10-00243-t009:** Recommendations for use of obstetric medication in COVID-19 patients.

Indication	Medication Class	Examples	Use in COVID-19	Precautions/Remarks
**Fetal maturation**	Corticoids	Betamethasone, Dexamethasone	Yes	Viral clearance of COVID19 may be delayed, though short-term treatment is assumed to be safe.
**Neuroprotection**	Membrane stabilising salt	Magnesium sulfate	Yes	Toxicity should be monitored (therapeutic range of 4.8 to 8.4 mg/dL OR 2.0 to 3.5 mmol/L). Magnesium is known to cause respiratory suppression. One of the first signs of toxicity is hyporeflexia.
**Tocolytic drugs**	Nonsteroidal anti-inflammatory Drugs	Indomethacine	No	NSAIDS increase the expression of ACE-2 receptors and are therefore not recommended in COVID19 patients.
	Calcium-antagonists	Nifedipine	Yes	No contraindications based on medication characteristics have been reported.
	Beta2-agonists	Salbutamol, Ritodrine	Preferably No	Risk of fluid overload by causing hypotension and fluid resuscitation.
	Oxytocin antagonist	Atosiban	Yes	No contraindications based on medication characteristics have been reported.
**Uterotonic drugs**	Prostaglandins	Prostaglandin E2, Misoprostol, Sulproston	Yes	No contraindications based on medication characteristics have been reported.
	Oxytocin receptor agonists	Oxytocin, Carbetocine	Yes	Risk of fluid overload because of inducing cardiovascular changes and ADH-like properties, especially when high doses or boluses. *
	Serotonergic, dopaminergic, α-adrenergic (ant)agonist	Methylergometrine	No	Risk of pulmonary edema has been reported, therefore use in COVID-19 patients is not recommended.
**Hemostatic drugs**	Inhibitor of trombolysis	Tranexamic acid	Yes	No contraindications based on medication characteristics have been reported.
**Vaccines**	Humoral immune response	Flu, Whooping cough	Yes	Flu and whooping cough can mimic COVID-19 infection, and are a risk factor for aggravating its severity
© ISIDOG COVID-19 2020 Guidelines				

* Oxytocin dosages estimated to be safe: Active third stage of labor: A bolus of 5 international units (IU) at time of delivery of the first shoulder. Uterine atony: A second slow bolus of 10IU oxytocin after 15 min or continuous oxytocin infusion of 10 IU/h in case (maximum of 60 IU/24 h). Abbreviations: ACE-2: angiotensin converting enzyme-2, ADH: antidiuretic hormone.

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
