# Peer review of "ISIDOG Recommendations Concerning COVID-19 and Pregnancy"

_diagnostics, 2020, doi:10.3390/diagnostics10040243_

Round 1

Reviewer 1 Report

The authors present a comprehensive set of recommendations concerning COVID-19 and pregnancy and including enumeration of specific measures to be taken in each individual patient according to her personal condition and disease severity. These recommendations are based on a broad consensus of specialists from different countries.

This paper represents a significant contribution to the field, it is well organized and comprehensively written. All adequate references to related and previous work are included.

There are quite a lot of typographical errors. I have pointed out some of them, but others may have escaped my attention. The paper should be re-checked in this respect.

In conclusion I recommend acceptance after minor revisions of the typos.

Typographical errors:

Page 1, line 19: ”manage a pregnant women” should read “manage pregnant women”

Page 1, line 21: “proven infected” should read “proven to be infected”

Page 1, line 21: “require special car” should read “require special care”

Page 1, line 29: “and ed on the following” shouls read “and on the following”

Page 2, Table 1, line 5: “muccoviscidose” should read “mucoviscidosis”

Page 2, Table 1, line 10: “Stadium 4” should read “Stage 4”

Page 3, line 94: “woman” should read “women”

Page 4, line 122: “woman” should read “women”

Page 4, line 131: “available of pregnancy” should read “available on pregnancy”

Page 4, Table 2, line 14: “Extreme premature deliverly” should read “Extreme premature delivery”

Page 7, line 246: “these woman had” should read “these women had”

Page 14, line 499: “pregnant woman tend” should read “pregnant women tend”

Author Response

Review 1

Typographical errors:

Page 1, line 19: ”manage a pregnant women” should read “manage pregnant women”

adjusted

Page 1, line 21: “proven infected” should read “proven to be infected”

adjusted

Page 1, line 21: “require special car” should read “require special care”

adjusted

Page 1, line 29: “and ed on the following” shouls read “and on the following”

adjusted

Page 2, Table 1, line 5: “muccoviscidose” should read “mucoviscidosis”

Adjusted in the definite table 1 in the text

Page 2, Table 1, line 10: “Stadium 4” should read “Stage 4”

Adjusted in the definite table 1 in the text

Page 3, line 94: “woman” should read “women”

adjusted

Page 4, line 122: “woman” should read “women”

adjusted

Page 4, line 131: “available of pregnancy” should read “available on pregnancy”

adjusted

Page 4, Table 2, line 14: “Extreme premature deliverly” should read “Extreme premature delivery”

Adjusted in the definite table 2 in the text, also corrected “hopsitalisation” to “hospitalization”

Page 7, line 246: “these woman had” should read “these women had”

adjusted

Page 14, line 499: “pregnant woman tend” should read “pregnant women tend”

adjusted

Reviewer 2 Report

This manuscript focuses on the clearly written recommendations concerning COVID-19 and pregnancy. Data are well presented.

These guidelines seem to be very useful in the age of SARS-CoV2 pandemic.

However, there are minor limitations:

What do you mean by:

-          “cfr table 1” in line 343;

-          “cfr supra” in line 366;

-          “weeks” in five lines of Table 3;

-          “s” in line 464;

-          “…” in line 502.

The statement about antiviral treatment (described in 535-537 lines) should be mentioned in the 10.2.1 part.

There are some abbreviations in the manuscript, which should be expanded (for example HCO3 in Table 6). On the other hand, the abbreviations once explained do not need to be translated again (for example MERS, SARS, PPE) and should be used throughout the article (for example ECMO in line 369).

The manuscript contains several typos and grammatical errors, affecting readability of the manuscript (for example “wich” in Table 1). Please proofread the manuscript carefully and if possible, have a native English speaker edit the manuscript.

Please use "Instruction for Authors" for all references.

Author Response

What do you mean by:

-          “cfr table 1” in line 343;

Has been adjusted to: “See Table 1 above”

-          “cfr supra” in line 366;

It has been adjusted to: “See above”

-          “weeks” in five lines of Table 3;

A part of Table 3 shifted, it will be replaced by the correct table shown below:

Table 3. Proposed out-patient pregnancy follow-up schedule in COVID-19 epidemic.

Table 3.

Proposed follow-up schedule for pregnant patients in COVID-19 epidemic.

Gestational age

Advised follow-up plan:

11-13+6

weeks

intake consultation documentation and risk stratification + blood type, complete blood count and serological testing (as per standard protocol) + clinical parameters + structural ultrasound scan (+/- trisomy screening)

20-22

weeks

clinical parameters + structural ultrasound scan + arranging appointment for glucose challenge test if indicated (as per country specific protocol) + instructions for alarm symptoms + at home follow up of blood pressure (BP) at 24 and 28 weeks

24-28

weeks

glucose challenge test ambulatory (without consultation) as per country specific protocol (general screening or risk selection)

30-32

weeks

clinical parameters + fetal biometry ultrasound scan + instructions for alarm symptoms + at home follow up of BP 2-weekly

34-36

weeks

clinical parameters + Group B streptococcal sampling (as per country specific protocol) + delivery planning
+ instructions for alarm symptoms + at home follow up of BP weekly

If a pregnant patient is positive for COVID-19 - routine consultations should be postponed by 14 days.
If a pregnant patient is assessed high risk and needs additional follow-up this needs to be assessed case by case.
We advise partners to be absent for routine consultations, to limit the exposure risk for health care providers.         
                                                                                                                                                                                                                  © ISIDOG COVID-19 2020 Guidelines

-          “s” in line 464;

A typing error, has been removed.

-          “…” in line 502.

Has been adjusted to “etc.”.

The statement about antiviral treatment (described in 535-537 lines) should be mentioned in the 10.2.1 part.

It has been added to the 10.2.1 part as well.

There are some abbreviations in the manuscript, which should be expanded (for example HCO3 in Table 6).

Explanations of abbreviations in tables have been added. The explanation “(Bicarbonate)” for HCO3 was added.

On the other hand, the abbreviations once explained do not need to be translated again (for example MERS, SARS, PPE) and should be used throughout the article (for example ECMO in line 369).

Has been corrected.

The manuscript contains several typos and grammatical errors, affecting readability of the manuscript (for example “wich” in Table 1). Please proofread the manuscript carefully and if possible, have a native English speaker edit the manuscript.

Please use "Instruction for Authors" for all references.

Has been corrected

English revision by native English speaker has been done.